

# Recognition of opinion leaders in blockchain-based social networks by structural information and content contribution

Chuansheng Wang, Xuecheng Tan and Fulei Shi

School of Management and Engineering, Capital University of Economics and Business, Beijing, China

## ABSTRACT

Exploring the influence of social network users in the blockchain environment and identifying opinion leaders can help understand the information dissemination characteristics of blockchain social networks, direct the discovery of quality content, and avoid the spread of rumors. Members of blockchain-based social networks are given new responsibilities by token awards and consensus voting, which alters how users connect to the network and engage with one another. Based on blockchain theory and the relevant theories of opinion leaders in social networks, this article combines structural information and content contributions to identify opinion leaders. Firstly, user influence indicators are defined from the perspective of network structure and behavioral characteristics of user contributions. Then, ECWM is constructed, which combines the entropy weight method and the criteria importance through intercriteria correlation (CRITIC) weighting method to address the correlation and diversity among indicators. Furthermore, an improved Technique for Order Preference by Similarity to Ideal Solution (TOPSIS), called ECWM-TOPSIS, is proposed to identify opinion leaders in blockchain social networks. Moreover, to verify the effectiveness of the method, we conducted a comparative analysis of the proposed algorithm on the blockchain social platform Steemit by using two different methods (voting score and forwarding rate). The results show that ECWM-TOPSIS produces significantly higher performance than other methods for all selected top N opinion leaders.

## INTRODUCTION

The centralized social networks have problems such as data privacy leakage and unrestricted spreading of fake news, which greatly violate user privacy security and affect their ability to discern information (*Cadwalladr & Graham-Harrison, 2018*). The unique decentralization, consensus mechanism, traceability and other features of blockchain can achieve a trustless and self-managed data interaction environment, which breaks through the technical bottleneck of existing social network management (*Hisseine, Chen & Yang, 2022*; *Sengupta et al., 2021*). With the combination of blockchain technology and social networks, several social media platforms have emerged, such as *Steemit*, *Lit*, *HyperSpace*, *Sapien*, *SocialX*,

Corresponding author
Fulei Shi, davidstoneman@163.com

*FORESTING* and *Minds* (*Guidi, 2020*). The decentralized features and economic incentives have changed the social behavior of users and the process of information dissemination (*Chen & Liu, 2022*). Therefore, the study of social network users in the blockchain environment has become a new issue of concern for relevant online public opinion departments and academics.

The blockchain social network allows users to achieve self-control of information, encourages users to generate high-quality content and spreads quality content by communicating with other users (*Zhan, Xiong & Xing, 2023*). Therefore, it is crucial to analyze the behavioral characteristics generated by users in the interaction process and identify influential users (*Panchendrarajan & Saxena, 2022*). In the process of information dissemination, influential opinion leaders have a great impact on other users and lead the public opinion (*Bamakan, Nurgaliev & Qu, 2019*). A great deal of attention has been paid to this branch. For example, *Jain (2022)* identified opinion leaders in social networks to control the spread of rumors in a timely manner and create a positive online environment. *Weng et al. (2023)* found that capturing the interaction between users and opinion leaders can accurately recommend information of interest to users and enhance information dissemination. *Lei, Zhang & Liu (2021)* analyzed the influence of opinion leaders on other users' behavior based on the characteristics of knowledge collaboration behavior in open-source projects (*OSPs*). *Wang et al. (2020)* analyzed that opinion leaders have more followers and influence on the information diffusion process. Therefore, identifying and analyzing the opinion leaders of the social networks in the blockchain environment helps to better understand the opinion dissemination characteristics of the blockchain social networks.

One of the common contexts of node importance is the identification of opinion leaders. By combining the local and global features of the network, we can determine which nodes have the most prominent positions. A rich body of literature has adopted a network-based approach to uncover these opinion leaders. (*Jain, Katarya & Sachdeva, 2020b*; *Yang, Rehman & Que, 2020*; *Maji, Mandal & Sen, 2020*). However, distinguishing from traditional social networks, the blockchain reputation-value system shifts users' attention to discovering quality content and finding trust (*Liu et al., 2022*). Meanwhile, the blockchain social networking platforms have added interactive features such as user reputation, voting, and bounties to change user characteristics and interaction behaviors (*Duan et al., 2020*). Under the influence of consensus mechanism (*Zheng & Boh, 2021*), irreversibility (*Yan et al., 2021*), and mechanism of valuable content (*Bai et al., 2021*), users who create and discover quality content and actively contribute to the community are more influential than those who have better network position (*Xie & Zhang, 2023*). Therefore, analyzing user characteristics and behaviors (*Girgin, 2021*) can better identify opinion leaders in blockchain social networks.

Consensus voting and incentives promote users to contribute quality content and avoid the spread of rumors. As members constantly participate in contributions and community activities, new characteristics are generated, which lead to changes in members' perceptions of the community and their influence (*Dong et al., 2020*). Therefore, exploring the characteristics of user contribution behaviors can better analyze user influence and identify

opinion leaders in the blockchain environment. This article proposes an ECWM-TOPSIS model, which combines both network structure and user contribution characteristics, for identifying blockchain social network opinion leaders. Firstly, we constructed social network user influence indicators in a blockchain environment. Secondly, considering the correlation between the user indicators to be evaluated and the decision makers' expectations, the ECWM-TOPSIS method was used to calculate the user's leadership score and rank the influence of social network users in the blockchain environment. Finally, through empirical analysis on the data of Steemit, the reliability of the proposed method for identifying opinion leaders is verified. Our contributions can be summarized as follows:

1.  In the blockchain social networks, users accumulate new characteristics through active contribution behaviors. Under the influence of self-interest or pro-social behaviors, these characteristics attract the attention of unfamiliar users, which can lead the topic direction by gaining more votes and expanding communication influence in topic discussions. Therefore, this article constructs a user influence indicator system based on structural information and content contribution dimensions to explore the cumulative effect of user influence.

2.  Considering the correlation between indicators and the expectation of decision makers, we proposed the ECWM-TOPSIS algorithm which combines the Entropy and CRITIC weighting methods to identify opinion leaders, avoiding the adverse impact of the combination of single weight and TOPSIS on the evaluation results.

3.  Experimental analysis is conducted by capturing data from Steemit, a mainstream blockchain social networking platform, on virtual currency opinion topics. In this study, we use voting scores and re-post rates to demonstrate the superiority of our proposed model over existing models.

The rest of this article is organized as follows. The background of opinion leader identification is listed in 'Related Work'. The proposed ECWM-TOPSIS algorithm is provided in 'Theoretical Basis'. The data collection and processing, weight assignment comparison and baseline algorithms, are given in 'The Proposed ECWM-TOPSIS Algorithm'. The results and analysis are presented in 'Experimental Setup'. Finally, the conclusion of this work is in 'Conclusions and Discussion'.

## RELATED WORK

In recent years, mounting studies have been conducted to identify opinion leaders for different network structures and user characteristics, such as centrality, reputation, and expertise. The selection of opinion leaders varies depending on the needs and environment. The social network in the blockchain environment is a network formed by the decentralized characteristics of the blockchain technology, and the characteristics of the social network still exist. For identifying and predicting social network influence nodes, social network analysis mainly relies on network structure (*Huang et al., 2017*). *Yang et al. (2019)* used entropy weight method to calculate the weight of each indicator and VIKOR method to identify key leader nodes. *Rehman et al. (2020)* measured the centrality of users and ranked opinion leaders based on betweenness centrality values. *Jain, Katarya & Sachdeva*

_(2020a)_ proposed a whale optimization algorithm (SNWOA) based on the network structure indicator optimization function to find the top $N$ opinion leaders by simulating the bubble-net hunting behavior of humpback whales. _Alexandre, Jai-sung Yoo & Murthy (2021)_ identified opinion leaders by measuring eigenvector centrality in combination with specific topics. _Rani & Kumar (2022)_ considered the heterogeneous interactions in the network and performed the identification and ranking of influential nodes using multi-criteria decision methods. _Yang et al. (2021)_ proposed a local centrality index of network nodes to identify influence nodes by combining network topology information, and the new algorithm has good performance. _Zhang et al. (2022)_ proposed a new multiple local attributes-weighted centrality ($LWC$) based on information entropy, combining degree and clustering coefficient to identify opinion leader.

Notably, most of centrality-based methods evaluate the influence of nodes from a certain perspective, which have limitations and can only be applied to some specific networks. In addition to network centrality, it was found that based on user characteristics (_Zhao et al., 2018_) can be used to identify opinion leaders, which can maximize the effectiveness of dissemination. _Jain & Sinha (2020)_ proposed the weighted correlated influence method to identify opinion leaders on microblogging platforms by combining relative influence based on timeline and user features. _Al-Emadi & Yahia (2020)_ identified opinion leaders on visual platforms, highlighting that features such as credibility and content quality helps to enhance user reputation and opinion leadership. _Yang et al. (2022)_ proposed a new influence calculation method to find local opinion leaders based on the semantic features of nodes. _Mao, Zhou & Xiong (2020)_ extracted relationships from user-posted content and constructs a weighted model from network structure and brand engagement to predict influencial users. Blockchain alters the way traditional volunteer-based social networks operate by facilitating members to provide quality content through consensus voting and incentives, as well as by fostering interpersonal communication and contributing behaviors that drive community growth (_Thelwall, 2018_). _Park (2023)_ from a value perspective, the more content a person shares, the more value he realizes, which enhances his influence in the collective. Therefore, analyzing the new features (_e.g._, reputation) generated by blockchain technology for social network users can better understand the information transfer and sharing behavior among members, and effectively identify opinion leaders in blockchain social networks (_Dal Mas et al., 2020_; _Tiscini et al., 2020_).

The research on opinion leaders in blockchain social networks is in its initial stage. Based on the theories related to blockchain and opinion leaders in social networks, this article builds an opinion leader identification model for blockchain social networks by establishing a user influence evaluation index system from two dimensions of network structure and user contribution characteristics.

# THEORETICAL BASIS

The network structure is well capable of predicting propagation influence. By analyzing the network structure information, users can prioritize those who are at the core of the topic and vote for quality content or generate dissemination behavior according to their

judgment. At the same time, in blockchain social networks, every transaction of users is recorded in the chain, and some new characteristics are accumulated by participating in community activities and providing quality content. These characteristics represent members' contributions to the community, and with the accumulation of contributions members are able to attract some strangers to pay attention to themselves, expand the scope of information dissemination and enhance their influence. Therefore, considering the impact of blockchain technology for social network users, the algorithm integrates the network structure and user contribution features to construct multiple influence metrics for analyzing the influence of users.

## Network structure

Online social networks are composed of a large number of users, each user can be considered as a node, and the edge is a channel for interaction between users. To simulate bidirectional propagation between users, a directed network $G = (V, E)$ is constructed in this article, where $V = V_1, V_2..., V_n$ denotes the set of all nodes; $E = e_1, e_2, \ldots, e_l$ denotes the set of connected edges of nodes. In this article, we use $X = (X_{ij}) n \times m$ to denote the adjacency matrix of G. It is obvious that $x_{ij} \neq x_{ji}$. In the directed network, if there is a connection between nodes, then $x_{ij} = 1$, otherwise $x_{ij} = 0$.

The degree of a node, as a local attribute ranking index, reflects the direct influence of a node on other nodes in the network, which also means that nodes with larger degrees will have more connections and information channels. In order to eliminate the influence of network size on degree centrality, the node degree centrality index is shown in Eq. (1):

$$DC(V_i) = \frac{1}{n-1} \sum_{j=1}^{n} x_{ij},$$ 

(1)

where in-degree centrality represents the number of comments, retweets and votes received by users of blockchain social media platforms from other users. Out-degree centrality represents the number of comments, retweets, and votes from users of the blockchain social media platform to other users.

The degree centrality does not consider the influence of neighboring nodes on the target node, nor does it consider the location of the target node in the whole network. Considering the importance of neighbor nodes can better reflect the influence of the target node, which can be measured by the eigenvector centrality. The eigenvector centrality index is shown in Eq. (2):

$$EC(V_i) = \lambda^{-1} \sum_{j=1}^{n} e_j x_{ij},$$ 

(2)

where $\lambda$ is the maximum eigenvalue of the adjacency matrix X; $e = (e_1, e_2, e_3, \ldots, e_n)^T$ is the eigenvector corresponding to the maximum eigenvalue of the adjacency matrix X. The importance of a node depends on the number of its neighboring nodes and on the importance of its neighboring nodes.

Betweenness centrality describes how a node controls or influences the relationship between two nodes that are not directly connected to it, reflecting the interface of the node

in the network, and can be used as an important indicator to control the information exchange in the network. The betweenness centrality can be measured by the number of times the shortest path between all node pairs passes through the target node. The betweenness centrality index is shown in Eq. (3):

$$BC(V_i) = \frac{2}{(n-1)(n-2)} \sum_{j=1, j \neq k}^{n} \frac{n_{jk}(V_i)}{n_{jk}}, \tag{3}$$

where $n_{jk}$ denotes the number of shortest paths between node $j$ and node $k$, $n_{jk}(V_i)$ denotes the number of shortest paths between node $j$ and node $k$ through node $i$. The larger betweenness centrality, the more it can control or influence the network relationship.

Similarly, from a global perspective, closeness centrality measures the influence of nodes in the network. This metric is mainly used to reflect the proximity between a node and other nodes in the network, and is measured by the sum of the shortest distance from a node to all other nodes. The closeness centrality index is shown in Eq. (4):

$$CC(V_i) = \left[ \frac{1}{n-1} \sum_{j=1, j \neq i}^{n} d_{ij} \right]^{-1}, \tag{4}$$

where $d_{ij}$ represents the shortest distance from node $i$ to node $j$. A higher value of the closeness centrality indicates that the node is more closely related to other nodes. The closeness centrality is more reflective of the global structure of the network than the degree centrality.

## User content contribution characteristics

Different from traditional social networks, blockchain's incentive mechanism encourages parties without trusting relationships to communicate and collaborate, and rewards or punishes users for their contribution behavior, which improves the quality of discovered content (*Lee & Ra, 2020*). At the same time, the more a person contributes to the blockchain social network, the higher the return and the higher the value realized, thus enhancing the influence in the community. Therefore, by analyzing the characteristics developed by users in the process of contribution can better understand the influence of users in the community. Considering that the blockchain social network is a community that distributes rewards through collective decisions, we try to measure the contribution characteristics of users in terms of recognition and user participation behavior.

Definition 1. Content Participation (*CP*)

User content engagement behaviors in blockchain social networks include posting and discovering quality content. By actively participating in the discussion of topics to attract more users' attention so as to enhance the influence. *Cai et al. (2018)* analyzed the reputation scoring mechanism in the white paper, which serves as an important way to measure the value members bring to the community, in that users accumulate reputation values by posting content that receives votes, leading to an increase in their standing in the community. When other members vote on posts, it increases the energy contributed by the user to participate in the community, and as the energy increases, the reputation increases.

In the Steemit, measuring user engagement behavior characteristics is calculated as shown in Eq. (5):

$$Reputation = (log(CP) - 9) * 9 + 25, \tag{5}$$

where, the CP value can be found in https://steemit.com/ under the corresponding username and the calculation formula can be found in the reputation section of the website (https://steemit.com/faq.html). The reputation of the user will be increased by the votes they get through the content participation. A higher reputation value will attract more users' attention and enhance their influence. In Fig. 1, we fit the curve based on the reputation evaluation mechanism, and it is easier to increase the reputation value in the initial stage. However, after accumulating to a certain level, the growth curve becomes slow, which means that more content engagement is needed to accumulate votes to keep the reputation value growing. Therefore, only by constantly engaging in content contributions can we improve our visibility, attract more people's attention, and thus increase our influence.

Definition 2. Reward ($RE$)

Every block created in the blockchain generates new tokens. Every day a fixed number of tokens are allocated to the network reward fund. These tokens are distributed to authors and curators for posting and voting on content. Emphasized the impact that token rewards can have on user behavior, particularly on the behaviors that shape the structure of social networks (*Ba, Zignani & Gaito, 2022*). The value of the reward is calculated as shown in Eq. (6):

$$R_i = M * \frac{V_i}{\sum_1^n V_n}, \tag{6}$$

where $R_i$ represents the amount of reward received for a single article, $M$ represents the total amount in the reward pool, $V_i$ indicates that the reward weight of the current article, and $\sum_1^n V_n$ represents the sum of the reward weights of all the articles sent on behalf of the prize.

With the continuous contribution of users, the status and roles shift, and the perception and values of individuals change, tending to take responsibility for the group and prioritizing the interests of the group when taking action (*Dong et al., 2020*). Therefore, we evaluate users influence in terms of originality presence follower/followings ratio and time to respond.

Definition 3. Originality Presence ($OP$)

$OP$ shows the number of original posts written by a user on a topic. Influential users tend to create content in their own words in topic discussions rather than reposting someone else's content most of the time. More original content can bring different perspectives to more people and avoid monotonous and boring content. In addition, the value of original presence is significantly much higher than re-posting works (*Park, 2023*). The value of $OP$ is calculated shown in Eq. (7):

$$OP_i = TA_i - OA_i, \tag{7}$$

**Peer**J Computer Science

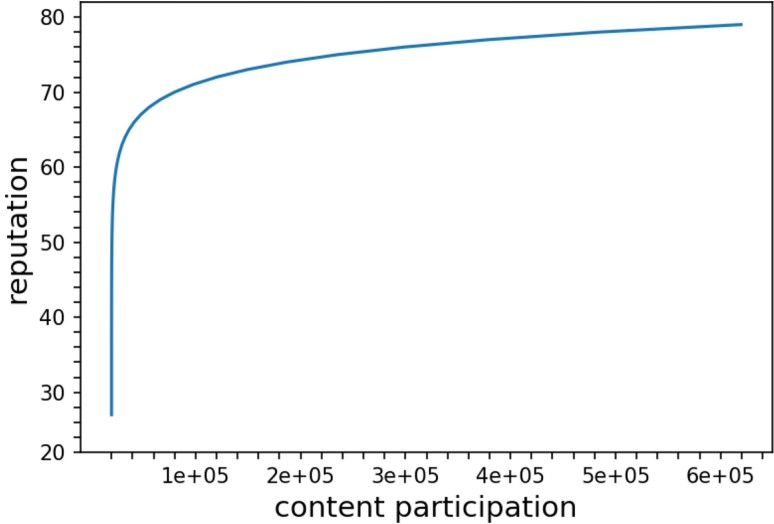

**Figure 1  User participation to accumulate reputation.**

where, $TA_i$ represents the total number of all posts made by user $i$ in the topic discussion and $OA_i$ represents the total number the posts made by user $i$ in the topic discussion were reposted.

Definition 4. Follower/following ratio($FFR$)

In Steemit, the social attributes and economic aspects of the user interact with each other. While Steemit emphasizes a reward system to motivate users, and it may seem that the richest users have more influence, some users may go on to increase their power by purchasing currency. This goes against the original design of the Steemit platform. Following others in the platform is free and may result in some financial reporting. However, attracting users in the platform is very difficult and requires a lot of effort. Therefore, user's follower/following ratio is proposed to analyze user influence (*Guidi, Michienzi & Ricci, 2020a*). The value of follower/following ratio is calculated shown in Eq. (8):

$$FFR_i = follower_i/following_i, \tag{8}$$

where $follower_i$ explains the number of follwers of user $i$ and the $following_i$ explains the number of followees of user $i$. For both users who did not follow others and users who were not followed by others, we set $FFR_i = 0$.

Definition 5. Time To Respond ($TTR$)

The purpose of blockchain social networks is to highlight quality content. When a topic is posted, users who post their opinions first and attract more votes are able to lead the trend of the topic, take control of the discourse in the topic, and increase their influence (*Jain, Katarya & Sachdeva, 2023*). Therefore, exploring the time users spend responding to a topic can be a useful way to analyze user influence. The $TTR$ value of user $i$ is calculated

as shown in Eq. (9):

$$TTR_i = \frac{1}{TF_i - PT},$$

(9)

where $TTR_i$ measures the extent of users' active response to the topic. $TTR_i$ is the time in minutes. $TF_i$ represents the time when user i first discussed the topic, and $PT$ represents the time when the topic was posted. To prove the responsiveness of the topic, an inverse approach is used.

## THE PROPOSED ECWM-TOPSIS ALGORITHM

This chapter introduces our proposed model. Traditional social networks assess influence by measuring individuals who are prominent in the network; however, features formed through contributed content in blockchain social networks can be voted on and recognized by more people as another underlying fact of user influence. Users can not only evaluate the content posted by neighboring users through structural information, but also select trusted users for direct communication and vote for their content based on the content contribution characteristics of other users in the chain. In Fig. 2, users build structural information features and user content contribution features to increase influence by posting content. The consensus mechanism broadcasts user features to the blockchain for other users to identify trusted users. Red nodes represent users with strong location structure, who post content that is passed to neighboring users to receive votes for rewards. However, there are also blue nodes that gain a high reputation by contributing to community content. These nodes are more likely to express their own views on the content and thus lead content trends. Therefore, we combine structural information and content contribution features to build a model to identify opinion leaders in blockchain social networks.

In blockchain social networks, each user can be regarded as an evaluation object, which can be comprehensively evaluated and ranked by attributes including network metrics and user-contributing features. TOPSIS (*Yang, Liu & Xu, 2018*), which synthesizes various evaluation indexes to find the best solution for multi-attribute decision-making problems, is one of the most effective strategies. Different evaluation attributes represent different characteristics of users, so it is necessary to assign reasonable weights to each attribute. In this article, we adopt the objective weight calculation method, which relies on the information presented by the data to determine the weights. A typical method is the EWM method (*Tang, Shi & Dong, 2019*). However, the EWM method is too sensitive to diversity, and an unrealistically high weight can be obtained if the value of a feature varies greatly. Therefore, in this article, we consider EWM and CRITIC (*Diakoulaki, Mavrotas & Papayannakis, 1995*) objective metrics to improve the weight assignment strategy and construct the ECWM-TOPSIS model. The influence of various aspects of blockchain social network users can be evaluated by our algorithm to identify opinion leaders in blockchain social networks. The algorithm consists of the following steps.

**Step1** Create a decision matrix. The network structure indicators (*e.g.*, DC, EC) and contribution characteristics indicators (*e.g.*, CP, RE) calculated in the previous section are selected to form a comprehensive evaluation index of user influence. In identifying

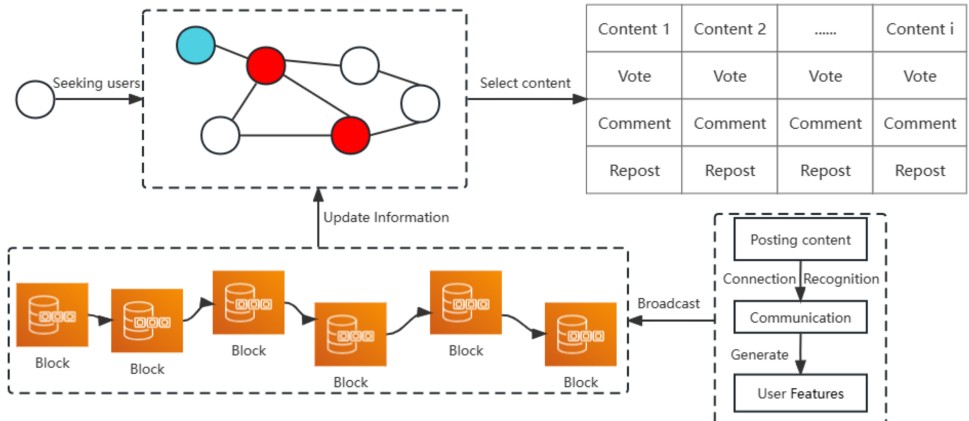

**Figure 2** **User interaction to shape network structure and user characteristics.**

opinion leaders, the number of users in the blockchain social network is selected as n, and the number of indicators is m to form the evaluation matrix X = ($X_{ij}$) $n \times m$, the decision matrix is shown in Eq. (10):

$$X = \begin{bmatrix} x_{11} & x_{12} & \cdots & x_{1m} \\ x_{21} & x_{22} & \cdots & x_{2m} \\ \vdots & \vdots & \ddots & \vdots \\ x_{n_1} & x_{n2} & \cdots & x_{nm} \end{bmatrix}, \tag{10}$$

where $X_{ij}$ denotes the $j_{th}$ evaluation indicator of user $i$.

**Step2** Standardized Decision Matrix. In order to eliminate the influence of the scale on the evaluation results, the indicators need to be standardized. The standardized equation is shown in Eq. (11):

$$x'_{ij} = \frac{x_{ij} - \min x_j}{\max x_j - \min x_j}, \tag{11}$$

$x'_{ij}$ and $x_{ij}$ denote the normalized and original values of the $j_{th}$ indicator for the $i_{th}$ user, max $x_j$ and min $x_j$ denote the maximum and minimum values of the $j_{th}$ indicator, respectively.

**Step3** Weight assignment (ECWM). In this article, we combine EWM and CRITIC to determine the weights. CRITIC determines the weights by utilizing the contrast strength and conflict nature of the criteria. It takes m metrics and n alternatives (blockchain social network users) as inputs and normalizes the value of each metric to the interval [0, 1] to compute the standard deviations. Based on the CRITIC weighting method, this study can construct the weighting formula as shown in Eq. (12):

$$W_j^1 = \frac{\sigma_j \sum_{k=1}^m (1 - r_{jk})}{\sum_{j=1}^m \left[ \sigma_j \sum_{k=1}^m (1 - r_{jk}) \right]}, \tag{12}$$

where, $W_j^1$ is the weight of the $j^{th}$ indicator obtained by the *CRITIC* method, $r_{jk}$ is the correlation coefficient between indicator $j$ and indicator $k$, and $\sigma_j$ is the standard deviation

of the measurement result of indicator $j$. The information entropy $E_j$ is calculated shown in Eq. (13), and the weight of each indicator is obtained shown in Eq. (14):

$$E_j = -\ln(n)^{-1} \sum_{i=1}^{n} p_{ij} \ln(p_{ij}) \tag{13}$$

$$W_j^2 = \frac{1 - E_j}{\sum_{j=1}^{n} (1 - E_j)}, \tag{14}$$

where n is the number of blockchain social network users and $p_{ij}$ is the probability of the occurrence of a particular indicator value. It is defined as $p_{ij} = x_{ij}/\sum_{i=1}^{n} x_{ij}$, where $x_{ij}$ is the value of $j_{th}$ indicator of the user $i$. And $W_j^2$ is the indicator weight calculated by the entropy weighting method. The CRITIC uses the standard deviation coefficient, mean squared deviation, and principal component analysis methods to comprehensively assess indicator comparison strength and data correlation. However, the entropy indirectly measures indicator diversity from dispersion degree. Therefore, we improve the combination of two assignment methods and constructs the ECWM method to achieve the complementary advantages between objective assignment methods. ECWM is calculated shown in Eq. (15):

$$W_j = \left( \alpha * W_j^1 + \beta * W_j^2 \right), \tag{15}$$

where $W_j$ is the weight of $j_{th}$ indicator using ECWM, $\alpha$ and $\beta$ are constants that represent the importance of Entropy and CRITIC method in calculating ECWM, and $\alpha + \beta = 1$. It is generally assumed that the two methods have equal status, i.e., $\alpha = \beta = 0.5$.

**Step4** Compute the weighted normalized matrix. In this step, the weighted normalized matrix $Z_{ij}$ is calculated by multiplying the normalized data by the corresponding weights: $Z_{ij} = x'_{ij} \times W_j$.

**Step5** Calculation of ECWM-TOPSIS based on Euclidean distance. Distance of positive ideal solution is shown in Eq. (16):

$$D_i^+ = \sqrt{\sum_{j=1}^{m} \left( Z_i^+ - Z_{ij} \right)^2}. \tag{16}$$

Distance of negative ideal solution is shown in Eq. (17):

$$D_i^- = \sqrt{\sum_{j=1}^{m} \left( Z_i^- - Z_{ij} \right)^2}, \tag{17}$$

where $Z_i^+ = (\max \langle z_{11}, z_{21}, \ldots, z_{n1} \rangle, \max \langle z_{12}, z_{22}, \ldots, z_{n2} \rangle, \ldots, \max \langle z_{1m}, z_{2m}, \ldots, z_{nm} \rangle)$, $Z_i^- = (\min \langle z_{11}, z_{21}, \ldots, z_{n1} \rangle, \min \langle z_{12}, z_{22}, \ldots, z_{n2} \rangle, \ldots, \min \langle z_{1m}, z_{2m}, \ldots, z_{nm} \rangle)$.

**Step6** Ranking of alternatives. The user influence is ranked according to their performance scores, which is shown in Eq. (18):

$$S_i = \frac{D_i^-}{D_i^+ + D_i^-}. \tag{18}$$

User lists are detected after each user's influence score is computed. The top $N$ users are selected as opinion leaders in descending order.

## EXPERIMENTAL SETUP

### Datasets

The Steemit (https://steemit.com/) is adopted for this study. Steemit blends blockchain technology and economic logic, and user-generated opinion data is permanently kept in the public blockchain. An economic system is designed to promote the content creation of information, which is a more typical online opinion platform under the blockchain environment. In this article, we select the virtual currency price with certain attention and influence as the topic. A web crawler is used to obtain data on Steemit, from which data in the time range from November 26, 2021 to June 16, 2022 are intercepted for analysis. First, the addresses of original opinion information on virtual currency price content are captured to form an initial address base. Secondly, based on the above addresses, field rules such as title author, post time, post content, bounty, number of retweets, retweeters, number of votes, voters, and number of posts were collected to further collect data. Finally, we obtain user reputation and other attribute information based on the author address database to form the original data samples, and use Excel, Spss and other data analysis software to clean and statistically analyze the collected data.

### Weight assignment comparison

In conducting multi-factor analysis, the determination of weights affects the outcome of the decision-making process. We use an objective combined weighting method that relies on the information presented by the data to determine the weights. Based on this, different methods are used to calculate the weights of each indicator as shown in Fig. 3.

In Fig. 3, the entropy weight method favors the attributes with higher diversity, *i.e.*, *RE*, *BC*, and *EC*. However, the CRITIC method calculates relatively lower values for *RE*, *BC*, and *EC*. Meanwhile, an extreme phenomenon can be observed, where entropy evaluates the minimum value globally for *CP* while CRITIC evaluates the maximum value globally for *CP*, and these two methods may result in irrational ranks.

Social networking in the blockchain environment eliminates the "centralization". Content creators have direct ownership of their content and interact directly with followers, fans, buyers, and other parties, with only smart contracts between them. Each independent user is free to choose trustworthy users by consensus mechanism, and jointly maintain community security and create content assets. Therefore, a reasonable increase in the value of *CP* and a decrease in the value of BC are more suitable for this network. ECWM combines the benefits of the entropy (EWM) and CRITIC methods and neutralizes their biases.

### Algorithms for comparison

To verify the performance of our proposed algorithm, two comparison methods, internal and external, are used to verify the results. In this article, we integrate the network structure and user content contribution characteristics. Therefore, in the internal comparison, we

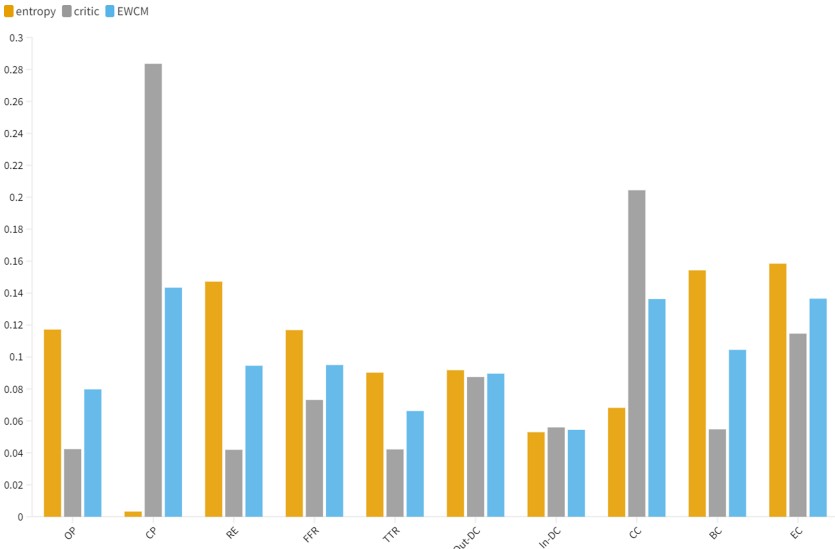

**Figure 3  Comparative analysis of indicator distribution.**

conducted a comparative analysis of network metrics as well as user feature separately to verify that the proposed model is superior to a single perspective. In the second comparison method, the existing algorithm entropy weighting based on Vlsekriterijumska Optimizacija I Kompromisno Resenje (EW-VIKOR) (*Yang et al., 2019*) and Steem Power (SP) in Steemit were used to evaluate the proposed algorithm. SP measures the voting influence and benefit distribution of users in the Steemit network. Similar to a stock, the more SP a user holds, the more influence the statements they make have on the quality of the message and therefore the more attention and agreement they will receive from other users (*Guidi, Michienzi & Ricci, 2020b*).

## EXPERIMENTAL RESULTS AND ANALYSIS

The evaluation of opinion leader detection is actually a difficult task because there is no standard method to evaluate opinion leaders. The focus of blockchain social networks is to highlight quality content to expand its dissemination and attract the attention of more users to gain votes on the content to increase the influence. Voting scores (*Hou, 2022*) are proven to be a ground truth for user influence to complement simulation-based communication capabilities. Meanwhile, *Alp & Öğüdücü (2019)* demonstrates that user propagation influence score can be measured by calculating re-post rate. Therefore, this study uses two measures, voting score and re-post rate, to assess the superiority of the proposed method.

In our proposed method, the top 30 users are shown in Table 1. It can be seen that the selected users have high reputation values, and further we observe the network structure score and content contribution score of each selected user. Dividing these users into two groups, those with high scores, such as vlemon, joelagbo, lupafilotaxia and tfame3868, can attract more users and thus establish a strong structural position by posting high

**Table 1  Influential users identification by ECWM-TOPSIS.**

| Rank | Users | Reputation | Network score | Contribution score |
|---|---|---|---|---|
| 1 | vlemon | 76 | 0.55 | 0.47 |
| 2 | joelagbo | 69 | 0.54 | 0.45 |
| 3 | lupafilotaxia | 72 | 0.53 | 0.46 |
| 4 | tfame3865 | 73 | 0.52 | 0.46 |
| 5 | josevas217 | 74 | 0.48 | 0.46 |
| 6 | gbenga | 73 | 0.45 | 0.46 |
| 7 | lebey1 | 65 | 0.47 | 0.43 |
| 8 | uoid | 53 | 0.5 | 0.38 |
| 9 | c3r34lk1ll3r | 60 | 0.44 | 0.46 |
| 10 | simonjay | 73 | 0.41 | 0.49 |
| 11 | lee2k | 75 | 0.29 | 0.58 |
| 12 | designieplay | 65 | 0.46 | 0.43 |
| 13 | joseph1956 | 68 | 0.43 | 0.44 |
| 14 | dani0661 | 62 | 0.43 | 0.41 |
| 15 | clixmoney | 76 | 0.37 | 0.47 |
| 16 | kingscrown | 81 | 0.24 | 0.55 |
| 17 | fun2learn | 70 | 0.39 | 0.45 |
| 18 | hardaeborla | 67 | 0.39 | 0.43 |
| 19 | samminator | 73 | 0.33 | 0.46 |
| 20 | reeta0119 | 73 | 0.32 | 0.46 |
| 21 | graduate | 56 | 0.26 | 0.51 |
| 22 | ahlawat | 75 | 0.32 | 0.47 |
| 23 | madridbg | 68 | 0.35 | 0.44 |
| 24 | abandi | 60 | 0.36 | 0.42 |
| 25 | emiliomoron | 69 | 0.34 | 0.44 |
| 26 | valchiz | 69 | 0.34 | 0.44 |
| 27 | winy | 61 | 0.38 | 0.41 |
| 28 | acom | 72 | 0.30 | 0.46 |
| 29 | benie111 | 67 | 0.32 | 0.43 |
| 30 | nainaztengra | 76 | 0.27 | 0.47 |

quality content. It means that they can get more support from other users in the network. The second type of users, who do not have high scores in one area, such as uoid, lee2k, kingscrow and graduate, are observed to rarely post content or post content that is overly homogeneous and difficult to attract the interest of other users. However, they still actively participate in discussions and are able to vote on content to highlight quality information, thus making their contribution to the community, which can also be part of an influential node.

## Performance evaluation

According to the consensus mechanism, the blockchain records each transaction and publishes the content to each user by broadcasting. In the blockchain social network, users

can vote on the content to choose the high value information to be delivered. Therefore, in this article, we measure the influence of users by analyzing the total number of votes they receive for content posted during the statistical time period, which we denote by $V_{\text{score}}$, calculated as shown in Eq. (19):

$$V_{\text{score}}(u_i) = \sum_{p=1}^{n} \text{vote}(u_{ip}), \tag{19}$$

where, $u_{ip}$ represents the $p_{th}$ post by user $i$, $\text{vote}(u_{ip})$ represents the number of votes received by user $i$ for the post. Users collect votes by posting content to gain support from more users. According to the features of blockchain social networks, content that receives more votes will be promoted to other users in priority. Therefore, influential users will always receive more votes, promote quality content and guide public opinion.

In addition to voting scores, re-post rates can also assess the influence of users. The blockchain social network aims to highlight quality content, avoid rumors, and direct public opinion. The re-post rate reflects other users' agreement with the user's opinion, which expands the dissemination range to increase the user's influence on the topic. Therefore, the influence of a user can be assessed by measuring the re-post rate, which is calculated as shown in Eq. (20):

$$Rp(u_i) = \begin{cases} r_{ij} / \sum_{p=1}^{n} i_p & \text{if } u_j \text{ reposted } u_i \\ 0 & \text{otherwise} \end{cases}, \tag{20}$$

where, $Rp(u_i)$ represents the re-post rate of user $i$, $\sum_{p=1}^{n} i_p$ represents the total number of posts of the user $i$ and $r_{ij}$ represents the total number of reposts received by user $i$ for all the posts. Reposting in blockchain social networks consumes certain resources, avoiding the abuse of reposting and the spread of rumors. Usually, rational users repost content posted by users with high influence to make quality content spread, so the influence of users can be assessed by analyzing the repost rate in blockchain social networks.

## Comparative analysis of algorithms in voting influence

In the Steemit, users are rewarded for voting on content they like. Voting on posts can be understood as an investment in shares, and once a post is published, it is rewarded based on the votes. For this reason, most users seek influencers to vote in the hope of receiving more rewards. We collected opinion information about virtual currency prices on the Steemit social platform to build the network. Based on the network, we built the network structure of users as well as the user contribution feature indicators, calculated the weights of the indicators, and ranked the users by our proposed method ECWM-TOPSIS. 1% (top 100) of users were selected as the candidate set of opinion leaders for the analysis of voting influence. In order to verify the validity of the model, the baseline model was constructed and tested under the same dataset. Users are ranked by the model to calculate the cumulative number of votes received by the top 100 users and compare it with our method. The voting influence is shown in Fig. 4.

With the increasing number of candidate sets, the ECWM-TOPSIS algorithm identifies a much higher number of users than the EW-VIKOR and other methods. The total

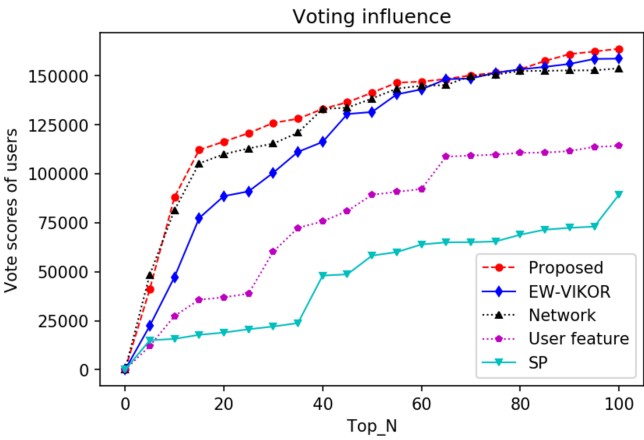

**Figure 4** **Voting of the top 100 by algorithm.**

number of votes for the top 100 users identified by our algorithm is 163,768, which is 76% of the total cumulative votes for all users, while the total number of votes for the users identified by the EW-VIKOR method is 158,720. In the top 20 opinion leader candidate sets, ECWM-TOPSIS rapidly increases the number of cumulative votes, far exceeding the other methods. Compared to the model that only considers network structure, our method is able to identify users that receive more votes, proving that the natural cumulative influence formed by combining user contribution features can compensate for the deficiency of only considering network contagion. Similarly, for the model considering only user contribution features, although it shows a growing trend with the increase of the candidate set, the cumulative votes tend to level off after the set of identified users is 60, and it is much smaller than that of our proposed method. It is worth mentioning that the SP curve only rises slowly with the increase of the candidate set. Further analysis suggests that this may be due to the fact that users with high SP are more involved in the act of voting on other users and seldom involved in the act of posting, and thus only receive a small number of votes. In order to verify the robustness of the model, we extract some data from the dataset for model validation. The results show that our proposed method is still ahead of other methods, rapidly improving in the top 20 candidate sets and obtaining the highest number of cumulative votes in recognizing the top 100 users.

## Comparative analysis of algorithms for spreading influence

In the above content we mentioned how to calculate the repost rates and analyze the spreading impact of different methods of identifying opinion leaders by comparing the repost rates. Similarly, we select the top 100 users as the candidate set of opinion leaders and analyze the total reposting rate accumulated by our proposed method in comparison with the total reposting rates accumulated by the users identified by the baseline method. In blockchain social networks, reposting consumes certain computational and storage resources, and users prefer to observe the topic's direction under the premise of uncertainty.

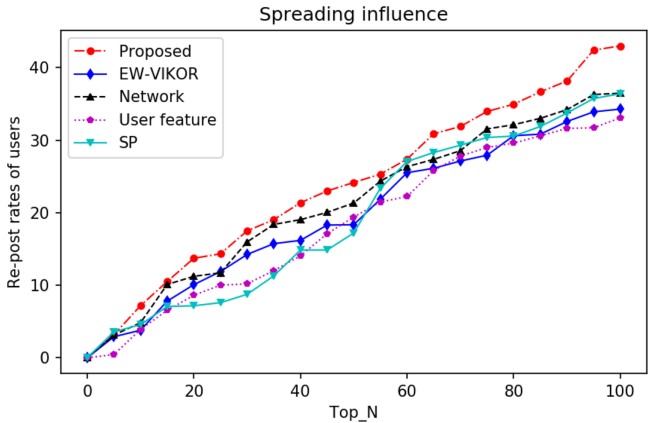

**Figure 5  Spread of the top 100 by algorithm.**

Users with high influence can attract more users to repost and promote the topic content. The spreading influence is shown in Fig. 5.

With the expansion of the selected datasets, all methods show an upward trend, among which the ECWM-TOPSIS method consistently leads and accumulates 43 in recognizing the top 100 users, outperforming the other methods. In identifying the top 20 users, the EW-VIKOR method shows nearly the same growth as our method, while the other methods show high and low growth trends. Comparative analysis of the network structure method and the user feature method reveals that both of them show a gradual stabilization as the number of identified users increases, while our method continues to grow, proving once again that considering both network structure and user contribution characteristics can effectively identify opinion leaders. As the topic is discussed, opinion leaders express their views to guide the topic orientation, and other users will choose to re-post the content if they agree with the views, thus allowing more users to see the quality content and expanding the spreading influence. SP, as the user-owned equity in Steemit, highlights the quality information by supporting others' content and represents the social influence of users. Observing the SP curve, we can see that it does not improve quickly in the initial stage, but instead, it improves rapidly in the later stages, and its re-post rate is 36.4 among the first 100 users, which is second only to our proposed algorithm. Similarly, we extract part of the data and repeat the above experimental process to compare and analyze the spreading influence of ECWM-TOPSIS with that of the baseline method to validate the robustness of the proposed method.

## CONCLUSIONS AND DISCUSSIONS

This article proposes a blockchain social network user opinion leader identification and influence analysis model to deepen the application of blockchain theory in social network opinion leader. This article combines network structure information and user contribution characteristics to construct blockchain social network opinion leader influence indicators and identify opinion leaders based on ECWM-TOPSIS method. The validity of our

proposed method is verified by analyzing the voting scores and repost rates, which can provide some reference and guidance for other blockchain social network platforms.

Our contributions can be summarized as follows: firstly, the two aspects of user network structure and contribution behavior characteristics can achieve better identification of opinion leaders; secondly, although users with more equity (SP) have more voting rights on blockchain social networks, they cannot accurately analyze the influence of opinion leaders and need to contribute more quality content to enhance their influence. Third, blockchain social network opinion leaders are generated with a certain degree of randomness, and it is possible to become an opinion leader by creating enough quality content.

However, there are still some shortcomings in this article. This article only establishes user influence evaluation indexes from network structure information and user contribution features, and it needs to capture more diverse information of blockchain social networks to improve the index system. A new ECWM method is proposed for this article, which uses EWM and CRITIC to assign the weights of indicators. Currently, the method gives the same importance to EWM and CRITIC to assign weights. As future work, we can explore different values of $\alpha$ and $\beta$ to see their impact on weight assignment. In addition, we only obtained data from the Steemit for comparative analysis of the models. In future work, we will explore different blockchain social platforms and build social networks to test the performance of our proposed method.

### Funding

This work was supported by the Beijing Municipal Universities Basic Research Funds in Capital University of Economics and Business (Grant No.XRZ2022029), the Beijing Municipal Education Commission Science and Technology general project (Grant No.KM202310038001) and the Natural Science Foundation of Beijing, China (Grant No. 20GLB028). The funders had no role in study design, data collection and analysis, decision to publish, or preparation of the manuscript.

### Grant Disclosures

The following grant information was disclosed by the authors:
The Beijing Municipal Universities Basic Research Funds in Capital University of Economics and Business: XRZ2022029.
The Beijing Municipal Education Commission Science and Technology general project: KM202310038001.
The Natural Science Foundation of Beijing, China: 20GLB028.

### Competing Interests

The authors declare that there are no competing interests.

## Author Contributions

- Chuansheng Wang conceived and designed the experiments, performed the experiments, prepared figures and/or tables, authored or reviewed drafts of the article, and approved the final draft.
- Xuecheng Tan conceived and designed the experiments, performed the experiments, analyzed the data, performed the computation work, prepared figures and/or tables, authored or reviewed drafts of the article, and approved the final draft.
- Fulei Shi conceived and designed the experiments, performed the experiments, performed the computation work, prepared figures and/or tables, and approved the final draft.

## Data Deposition

The data is available at Zenodo: (2023). Recognition of opinion leaders in blockchain-based social networks by structural information and content contribution. https://doi.org/10.5281/zenodo.7654286.

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
