# Peer review of "Recognition of opinion leaders in blockchain-based social networks by structural information and content contribution"

_PeerJ Computer Science, doi:10.7717/peerj-cs.1549_

## Round 0.1 · original submission · Major Revisions

Dear Authors,

Your submission has been reviewed by two referees who both raised serious concerns about the methodology, design and how the conclusions are drawn and validated. The recommendation is 'major revisions' - please consider all the comments of the Referees carefully and be aware of the extent and depth of the extra work required to bring the paper to a satisfactory level for publication.

Reviewer 1 ·

Basic reporting

Refer to the pdf attached.

Experimental design

Refer to the pdf attached.

Validity of the findings

Refer to the pdf attached.

Additional comments

no comment

Annotated reviews are not available for download in order to protect the identity of reviewers who chose to remain anonymous.

Reviewer 2 ·

Basic reporting

The following sentence “The existing social network opinion leader identification models cannot 121 be effectively applied to blockchain social networks” should be properly motivated.

In Network Structure section, the authors define a directed network as G = (V,E,A). The matrix A contains information on G, so I cannot see why A is an element of G. In this section, authors provide definitions of Degree centrality which require to be formalised in mathematical terms. Also, some definitions sound poorly compelling.

In a social platform, influencers receive high attention from the community of users, usually measured through likes, and other consensus-based indicators. Here, the authors state that the number of likes a post receives quantifies its contribution to a platform. That would imply that users publishing many posts, receiving only a few likes, contribute less than users publishing a single post receiving many likes. If so, that should be properly motivated.

How the numerical parameters in (1), i.e. 9s, have been defined?

What does energy mean in this context?

What is the definition of ‘original post’?

What is the FFR value for users that follow no users?


In equation (5), the parameter TTR has an index ‘i’, that is not present in the text.

In general, the section related to parameters and quantities needs strong revision.


In the section, “The proposed ECWM-TOPSIS Algorithm” matrix A has a different definition/meaning than the one given in the previous section.

In line 245, W has only an upper index.


Figures have a very poor caption.

What is the CRITIC method?


Equations in the remaining sections of the manuscript need revision. For instance, u’ \in |voting(u_i)| typically means that a given element belongs to a set.

In the section conclusion, statements such as ‘existing algorithms do not properly consider the characteristic of social network users …’ are poorly motivated.

Experimental design

no comment

Validity of the findings

comment in part (1)

---

## Round 0.2 · Major Revisions

Dear authors,

Thank you for submitting a revised version of the paper.

Referee 1 expressed overall satisfaction with the revisions, as you have addressed all points previously mentioned by both reviewers, increasing the confidence in the soundness of your approach.

However they also informed me that the clarity of the article would still benefit from further restructuring and editing.

I agree with the concerns raised to me by Referee 1and so I would like to give you a further opportunity to refine the presentation and overall clarity of your manuscript. Please see specific comments below:

(I) The paper would benefit from a professional proofreading to increase the clarity and conciseness of English sentences.

(ii) In line 66, please check the reference to Batinic (2016) as the referee could not find any compelling link with the argument presented by the authors.

(iv) Line 283 (and in general throughout the main text) add references to original papers introducing methods such as EWM, CRITIC, etc. used in the current approach.

(v) In Section “THE PROPOSED ECWM-TOPSIS ALGORITHM”, when explaining the sub-steps please clarify and stress again in the context of the blockchain social networks with simple examples what the different variables would correspond to. E.g., indicate explicitly what the indicator of a node would correspond to. Also please provide an intuition for the proposed method in terms of selection of top-nodes. 


(v) The sections "Comparative analysis of algorithms in voting influence" and "Comparative analysis of algorithms for spreading influence" should include further details on how the experiments were conducted and compared, a quantitative estimate of the gain of the proposed algorithm and possible limitations of the analysis (i.e. are these increases in performance statistically robust upon repetition of the experiments?).

(vi) In terms of limitations please also expand and clarify the statement in lines 467-469.

In general please make sure that the description of your methods and experiments are given with sufficient details to be fully reproducible in future studies.

Reviewer 1 ·

Basic reporting

no comment

Experimental design

no comment

Validity of the findings

no comment

Additional comments

no comment

---

## Round 0.3 · accepted · Accept

I feel that the authors have satisfactorily addressed the requested revisions, and the paper is now ready to appear on the journal.